# A Degradation Motif in STAU1 Defines a Novel Family of Proteins Involved in Inflammation

**DOI:** 10.3390/ijms231911588

**Published:** 2022-09-30

**Authors:** Yulemi Gonzalez Quesada, Luc DesGroseillers

**Affiliations:** Département de Biochimie et Médecine Moléculaire, Faculté de Médecine, Université de Montréal, 2900 Édouard Montpetit, Montréal, QC H3T 1J4, Canada

**Keywords:** Staufen1, inflammation, protein degradation, E3 ubiquitin ligase, degron, TRIM25

## Abstract

Cancer development is regulated by inflammation. Staufen1 (STAU1) is an RNA-binding protein whose expression level is critical in cancer cells as it is related to cell proliferation or cell death. STAU1 protein levels are downregulated during mitosis due to its degradation by the E3 ubiquitin ligase anaphase-promoting complex/cyclosome (APC/C). In this paper, we map the molecular determinant involved in STAU1 degradation to amino acids 38–50, and by alanine scanning, we shorten the motif to F^39^PxPxxLxxxxL^50^ (FPL-motif). Mutation of the FPL-motif prevents STAU1 degradation by APC/C. Interestingly, a search in databases reveals that the FPL-motif is shared by 15 additional proteins, most of them being involved in inflammation. We show that one of these proteins, MAP4K1, is indeed degraded via the FPL-motif; however, it is not a target of APC/C. Using proximity labeling with STAU1, we identify TRIM25, an E3 ubiquitin ligase involved in the innate immune response and interferon production, as responsible for STAU1 and MAP4K1 degradation, dependent on the FPL-motif. These results are consistent with previous studies that linked STAU1 to cancer-induced inflammation and identified a novel degradation motif that likely coordinates a novel family of proteins involved in inflammation. Data are available via ProteomeXchange with the identifier PXD036675.

## 1. Introduction

Most recent studies reveal a connection between inflammation and cancer [1]. Indeed, tumor-promoting inflammation is a characteristic of cancer [2]. Reciprocal interactions between cancer cells and inflammatory cells create an inflammatory tumor environment that predisposes to tumor progression, metastases, and recurrence. Both cancer and inflammation are complex processes, and the discovery of new inflammation-related pathways in relation to cancer may lead to important advances in the development of therapies against cancer.

The success of cell functions is dependent on the control of protein levels through a coordinate process that balances translation and degradation [3]. Dysregulation of this equilibrium has disastrous consequences on cell homeostasis [4]. Protein degradation is mostly mediated by the ubiquitin–proteasome system (UPS) [5]. In this process, ubiquitin is transferred to target proteins via a three-enzyme process that relies on the activating enzyme (E1), ubiquitin-conjugating enzyme (E2), and ubiquitin ligase (E3). Monoubiquitinated, polymonoubiquitinated, and polyubiquitinated proteins can then be processed by the 20S, 26S, or 30S proteasomes for degradation [6,7,8]. Polyubiquitinated proteins targeted for degradation contain polyubiquitin chains in which ubiquitin monomers are covalently added on lysine 48 (K48-linked) or lysine 11 (K11-linked). However, other types of polyubiquitination have alternative consequences on proteins. For example, K63-linked polyubiquitin chains do not tag proteins for degradation but rather form a platform for new protein interactions and activation [9]. The ubiquitin system plays important roles in the control of multiple processes, including cell proliferation, signal transduction, transcriptional and translational regulation, cell death, and immune response [5]. It is required for the activation of inflammatory pathways and for their attenuation to prevent tissue damage [10]. E3 ubiquitin ligases are thus considered as targets for inflammatory diseases [11].

Humans have an estimated 700 E3 ubiquitin ligases [12] that specifically ensure the last step of ubiquitination on target proteins. They usually recognize a short amino acid sequence consensus motif, known as degron, on target proteins. Target proteins can contain multiple degrons. However, only few E3 ligases have known degron motifs [13], hampering our capability to identify families of target proteins that are coordinately regulated by protein degradation in response to changing physiological environments.

The E3 ubiquitin ligase anaphase-promoting complex/cyclosome (APC/C) controls cell cycle events through the degradation of selected proteins [12,14]. Two coactivators, cell division cycle 20 (CDC20) and fizzy and cell division cycle 20 related 1 (CDH1), allow APC/C to target a large range of proteins and to control several steps of the cell cycle [15]. Via its coactivators, APC/C can recognize several consensus degrons [16], the most characterized ones being the D-box [17] and KEN-box [18] motifs. One of the targeted proteins is the RNA-binding protein Staufen1 (STAU1) [19]. STAU1 is a double-stranded RNA (dsRNA)-binding protein that is expressed as two different isoforms of 55 and 63 kDa [20,21], the 55 kDa isoform being the most highly expressed. STAU1^63^ is identical to STAU1^55^ but carries an 81-amino-acid extension at its N-terminal extremity. STAU1^55^ contains four dsRNA-binding domains (RBD), two of them, RBD3 and RBD4, being able to bind dsRNA in vitro [20,21,22]. Accordingly, STAU1^55^ is a post-transcriptional regulator that controls the fate of bound RNAs, from localization, transport, translation, and alternative splicing to degradation [23,24,25,26,27,28]. STAU1^55^ expression is essential in untransformed cells as it facilitates checkpoint transition [29]. STAU1^55^ is involved in multiple pathways that control development, differentiation, stress granule regulation, synaptic plasticity and long-term potentiation, apoptosis, cell proliferation, RNA virus replication, and immune response to virus infection [23,30,31,32,33]. It is also clearly linked to cancer development, cell migration, metastasis, and tumor angiogenesis [23]. In cancer cells, STAU1^55^ overexpression impairs cell proliferation and induces apoptosis [19,33]. Interestingly, recent studies link STAU1^55^ to proinflammatory processes [32,34].

STAU1^55^ protein levels vary during the cell cycle, being high in S/G_2_ and decreasing during mitosis as a result of protein degradation via APC/C [19]. Interestingly, STAU1^55^ association with CDC20 or CDH1 depends on a molecular determinant located in the first 88 N-terminal amino acids, a domain known as RBD2 [19]. In this paper, we report the characterization of the STAU1^55^ degradation motif targeted by the APC/C (FPxPxxLxxxxL-FPL motif) and show that it is shared by 15 additional proteins. Strikingly, 11 of them are linked to inflammation. We further show that the E3 ubiquitin ligase TRIM25 recognizes the motif for protein degradation. This protein is involved in the innate immune response and in interferon production following viral infection. These results suggest that STAU1^55^ and other proteins with the same degradation motif can be regulated by TRIM25 in a context of inflammation to trigger and/or attenuate the immune response.

## 2. Results

### 2.1. RBD2 Is Sufficient to Confer Protein Degradation Capacity

STAU1^55^ was reported to be a substrate of the E3 ubiquitin ligase APC/C and, thus, to be polyubiquitinated and degraded during mitosis [19]. The motif responsible for STAU1^55^ degradation likely resides within RBD2 since its deletion prevents STAU1^55^ interaction with CDH1 and CDC20, two coactivators of APC/C, and its degradation by the proteasome [19]. To determine whether RBD2 is sufficient to confer protein degradation, the first 88 amino acids of STAU1^55^ (RBD2) were fused to YFP, a protein that is normally not degraded by the 26S proteasome. Then, we compared the levels of RBD2-YFP with that of STAU1^55^-YFP and STAU1^Δ88^-YFP (Figure 1A) in the presence or absence of the proteasome inhibitor MG132 (Figure 1B). As expected [19], STAU1^55^-YFP was stabilized in the presence of MG132, while STAU1^Δ88^-YFP was not. Similarly, RBD2-YFP was stabilized following proteasome inhibition, while YFP was not, indicating that the first 88 amino acids of STAU1^55^ are necessary and sufficient to confer protein degradation capacity.

To determine whether RBD2 is sufficient to be recognized by CDH1, plasmids coding for HA-CDH1 and either STAU1^55^-YFP, STAU1^Δ88^-YFP, RBD2-YFP, or YFP were cotransfected in HEK293T cells, and the amount of the proteins quantified by Western blotting (Figure 1C). Stau1^55^-YFP and RBD2-YFP levels decreased when expressed in the presence of HA-CDH1, whereas STAU1^Δ88^-YFP and YFP levels were stable. All these results confirm that RBD2 contains a motif responsible for STAU1^55^ degradation.

### 2.2. Amino Acids 38 to 51 Are Involved in STAU1^55^ Stabilization and Interaction with the APC/C Coactivator CDH1

To define the amino acid sequence involved in STAU1^55^ interaction with CDH1 and in STAU1^55^ degradation, progressive deletions were introduced at the N-terminal end of STAU1^55^ (Figure 2A). These mutants were expressed in HEK293T cells in the presence or absence of MG132, and their levels of expression quantified by Western blotting (Figure 2B). STAU1^55^-HA_3_ and STAU1^Δ88^-HA_3_ were used as positive and negative controls, respectively. Deletion of the first 88 or 60 amino acids prevented STAU1^55^ stabilization upon MG132 treatment, indicating that residues 60 to 88 do not contain the degradation motif. In contrast, proteins with deletions of 7, 17, 25, or 37 amino acids were stabilized in cells treated with MG132 compared with cells treated with DMSO, indicating the presence of a degradation motif between amino acids 38 and 60. A mutant with a deletion of 46 amino acids showed variable, although weak, stabilization, suggesting that the deletion may overlap the degradation motif (Figure 2B).

These results were confirmed in the context of RBD2. Deletions of 37, 46, and 51 amino acids were generated at the N-terminal end of RBD2-YFP (Figure 2C), and the resulting proteins were expressed in HEK293T cells in the presence or absence of MG132 (Figure 2D). A mutant lacking the first 37 amino acids was stabilized in the presence of MG132, as observed with mutants of the full-length protein. In contrast, the deletion of the first 46 or 51 residues of RBD2 did not cause a significant stabilization of the mutant proteins in the presence of MG132 suggesting that the degradation motif can be narrowed down to amino acids 38–51 of STAU1^55^. Plasmids coding for the mutant RBD2 proteins were also c-transfected with a plasmid coding for HA-CDH1 (Figure 2E). The expression levels of mutants with the deletion of 37 amino acids decreased in the presence of CDH1, similar to the decrease in the full-length STAU1^55^ protein. In contrast, the deletion of 51 amino acids prevented protein stabilization.

### 2.3. Amino Acids 38 to 60 Are Involved in Ubiquitin K11-Dependent Ubiquitination

Substrates to be degraded following ubiquitination by the APC/C coactivator CDH1 are enriched in ubiquitin K11 chains. Therefore, the K11-dependent ubiquitination of STAU1^55^ proteins and mutants was tested by Western blotting following the coexpression of GFP-ubi or GFP-ubi-K11R (Appendix A). In the presence of GFP-ubi, STAU1^55^-HA_3_ and STAU1^Δ37^-HA_3_ were polyubiquitinated, STAU1^Δ46^-HA_3_ was only lightly ubiquitinated, while STAU1^Δ88^-HA_3_ and STAU1^Δ60^-HA_3_ were not (Appendix A), consistent with their degradation profiles (Figure 2B). In contrast, when cotransfected with a plasmid coding for ubiquitin K11R, polyubiquitination was highly reduced on all proteins (Appendix A). This result supports previous data that STAU1^55^ degradation depends on the APC/C coactivator CDH1. Interestingly, the incomplete loss of STAU1^55^ polyubiquitination when using ubi-K11R suggests that STAU1^55^ also carries CDH1-dependent ubi-K48 chains or that it can be targeted by other E3 ubiquitin ligases.

### 2.4. Alanine Scanning of Amino Acids 37–51 Identifies a Novel Motif of Protein Degradation

To define the degradation motif in RBD2, we generated STAU1^55^ mutants using the alanine substitution of each amino acid between 37 and 51. We tested the levels of expression of each mutant, in the presence or absence of MG132, by Western blotting (Figure 3). Results indicated that five specific amino acids are involved in STAU1^55^ degradation. The degradation motif of STAU1^55^ was identified as F^39^P-x-P-x(2)-L-x(4)-L, named FPL-motif, where the contribution of each residue seems to be equally important since the alteration of any of them abrogates the stability of the protein.

### 2.5. STAU1^55^ Degradation Motif Defines a Family of Proteins Involved in Inflammation

To determine whether the novel degradation motif of STAU1^55^ can be found in other proteins, we screened the PROSITE database with the motif sequence and identified 15 additional proteins with a similar motif (Table 1). None of them were RNA-binding proteins or linked to cell cycle regulation. However, 12 out of the 16 proteins were previously linked to inflammation and/or immune response.

### 2.6. The Degradation Motif Is Involved in the Degradation of MAP4K1 by the UPS

To determine whether the degradation motif found in STAU1^55^ is also functional for the control of the steady-state levels of other proteins that carry the same motif, we first expressed MAP4K1-HA_3_ in HEK293T cells in the presence or absence of MG132 and quantified its level of expression by Western blotting (Figure 4). In the presence of MG132, a 4.24 ± 1.14-fold increase was observed, indicating that MAP4K1-HA_3_ is normally degraded by the UPS. Then, a point mutation (F651A) was introduced in the putative degradation motif to determine whether the FPL-motif is relevant for MAP4K1 protein degradation. The F651A mutant showed only a modest level of stabilization (1.70 ± 0.15) (Figure 4), revealing that the motif is relevant for the degradation of the protein. These results straighten the possibility that the FPL-motif defines a family of proteins involved in inflammatory processes.

### 2.7. CDH1 and CDC20 Recognize the STAU1^55^ FPL-Motif but Are Not Involved in MAP4K1 Degradation

To determine whether APC/C degrades STAU1^55^ via the FPL-motif, plasmids coding for STAU1^55^-HA_3_ and STAU1^F39A^-HA_3_ were cotransfected with plasmids coding for HA-CDH1 or HA-CDC20 in HEK293T cells. As expected, STAU1^55^-HA_3_ was destabilized by HA-CDH1 and HA-CDC20 (Figure 5A). In contrast, STAU1^F39A^-HA_3_ was not (Figure 5A), indicating that APC/C-mediated STAU1^55^ degradation requires the FPL-motif. To determine whether APC/C also recognizes the FPL-motif in MAP4K1, we cotransfected plasmids coding for MAP4K1-HA_3_ or MAP4K1^F651A^-HA_3_ with or without plasmids coding for either HA-CDH1 or HA-CDC20 in HEK293 cells and determined their levels of expression (Figure 5B). Unexpectedly, the amounts of MAP4K1-HA_3_ or its mutant were not affected by HA-CDH1 or HA-CDC20 expression, indicating that these proteins are not involved in MAP4K1 degradation. This result also indicates that other E3 ubiquitin ligases are involved in the control of MAP4K1 expression.

### 2.8. BioID2/TurboID Identifies Putative E3 Ubiquitin Ligases Involved in STAU1^55^ Degradation

We then used the biotin ligase strategy to identify E3 ubiquitin ligase(s) that interact(s) with STAU1^55^ via the degradation motif. To increase the probability of identifying the E3 ligase, we biotin-labeled proteins in the presence of the STAU1^55^–biotin ligase fusion protein for either 16 h in asynchronous hTERT-RPE1 cells (BioID2) or 3 h in cells synchronized in mitosis or in G_1_/S phase transition (TurboID). Labeled proteins were pulled down with streptavidin-coated magnetic beads and analyzed by mass spectrometry (Appendix A). Due to the difficulty of generating an optimal negative control with these techniques, we only kept proteins with a total spectrum enriched at least four times compared with controls for further analysis. As expected, proteins known to be linked to STAU1^55^ (e.g., UPF1, PP1, FXR1, STAU2) were found in the proximity of STAU1^55^ and were grouped in GO categories that matched known STAU1^55^ functions (e.g., translation, RNA degradation, ribonucleoprotein complex assembly) (Appendix A), indicating the specificity of the results. While CDH1 was not found in the proximity of STAU1^55^, CDC20 was found in its proximity in cells synchronized in mitosis, in agreement with previous data showing that STAU1^55^ is a target of APC/C [19]. Interestingly, three additional E3 ubiquitin ligases, TRIM25, TRIP12, and ZNF598, were repeatedly found in the proximity of STAU1^55^ with significant peptide counts (Table 2).

### 2.9. The E3 Ubiquitin Ligase TRIM25 Is Involved in STAU1^55^ and MAP4K1 Degradation

To determine whether TRIM25, TRIP12, or ZNF598 could be involved in STAU1^55^ degradation, we downregulated the expression of the three proteins with dsiRNAs and compared the amounts of STAU1^55^ in control and siRNA-expressing cells (Figure 6). RT-qPCR experiments confirmed that the E3 ubiquitin ligases were downregulated by the expression of dsiRNAs (Figure 6A). Following the depletion of TRIM25, the amounts of STAU1^55^ were increased (Figure 6B), indicating that TRIM25 can degrade STAU1^55^. In contrast, TRIP12 and ZNF598 depletion had no effect on STAU1^55^ stability. We then tested whether TRIM25 is involved in MAP4K1-HA_3_ degradation. In the presence of dsiRNA against TRIM25, the amounts of MAP4K1-HA_3_ was increased, indicating that it is a target of TRIM25 (Figure 6C). These results indicate that the E3 ubiquitin ligase TRIM25 is implicated in the degradation of STAU1^55^ and MAP4K1.

### 2.10. TRIM25 Recognizes the Degradation Motif

To determine whether TRIM25 uses the degradation motif to target the proteins to the proteasome, we expressed STAU1^F39A^-HA_3_ (Figure 6D) and MAP4K1^F651A^-HA_3_ (Figure 6E) proteins in the presence or absence of dsiRNAs against TRIM25. We first confirmed the depletion of TRIM25 by RT-qPCR (left panels). The quantification of the amount of the mutant proteins (right panels) revealed that they were not affected by TRIM25 depletion, indicating that the mutation in the FPL-motif prevents their degradation by TRIM25.

## 3. Discussion

In this paper, we identified the molecular determinant that causes STAU1^55^ polyubiquitination and degradation by the 26S proteasome degradation following its interaction with the E3 ligase APC/C coactivators, CDC20 and CDH1. The FPL-motif is not related to previously identified targets of the APC/C coactivators, such as the D-box [17] or KEN-box [18] motifs. Interestingly, the FPL-motif is found in 15 additional proteins and shown to be required for the degradation of MAP4K1. Nevertheless, MAP4K1 is not a target of APC/C, indicating that additional E3 ligase(s) can also recognize the FPL-motif. Interestingly, a search for other E3 ubiquitin ligases that target both STAU1^55^ and MAP4K1 identifies TRIM25, a protein involved in inflammation and immune response [48]. Strikingly, STAU1^55^ [31,32,49,50] and most of the proteins that share that FPL-motif (see below) were previously associated with inflammation and/or immune response, suggesting that the motif is part of a signaling pathway that coordinately responds to inflammation. Our results indicate that the degradation motif of STAU1^55^ is a target site for at least two E3 ubiquitin ligases that lead to STAU1^55^ degradation under different cell signaling pathways, cell cycle (APC/C) and inflammation (TRIM25).

### 3.1. STAU1^55^ Degradation Motif Is a Noncanonical Sequence Targeted by APC/C

We previously showed that STAU1^55^ is polyubiquitinated by the E3 ubiquitin ligase APC/C and, subsequently, degraded by the proteasome [19]. Both APC/C coactivators, CDH1 and CDC20, were found to specifically interact with STAU1^55^. Surprisingly, the co-activators did not interact with the D-box motif at the C-terminal end of the protein but rather with the RBD2 domain. RBD2 is composed by the first 88 residues at the N-terminal end of STAU1^55^. Its sequence is similar to a double-stranded RNA-binding consensus motif, although it has no detectable RNA-binding activity in vitro. Accordingly, it is expected to adopt the known dsRBD tridimensional structure α_1_β_1_β_2_β_3_α_2_ [51]. Computer modeling of the protein structure suggests a disordered β1 sequence while the α_1_β_2_β_3_α_2_ structures are conserved (Appendix A). The first four amino acids of the FPL-motif are located in the loop between the disordered region and β-sheath 2, whereas Leu^50^ is found within β2. Therefore, the FPL-motif is likely exposed to putative external ligands. The FPL-motif is novel and was not previously described. It is necessary and sufficient for protein degradation since it induces the degradation of proteins on which it is fused. Nevertheless, not all FPL-containing proteins are targeted by APC/C, as observed for the MAP4K1 protein (Figure 5). It is possible that their different subcellular localization accounts for this observation, APC/C being a nuclear protein and MAP4K1 a membrane-associated protein.

The FPL-motif may allow a flexibility or specificity for the degradation of STAU1^55^ by APC/C that is spatially and/or timely different from that observed for other APC/C target proteins with D-box or KEN-box motifs (see also below). Indeed, APC/C was shown to recognize multiple degrons [16], suggesting that it may differentially coordinate the expression of subfamilies of proteins. Interestingly, the FPL-motif is well conserved among STAU1 proteins in mammals (Appendix A) but not in nonmammalian species, suggesting that mammalian STAU1 proteins have recently evolved and acquired novel functions associated with the FPL-motif. It is also conserved in MAP4K1 (Appendix A) and other FPL-motif-containing proteins (Appendix A) in mammals but not in nonmammalian proteins (not shown).

### 3.2. The FPL-Motif Is Shared by a Family of Proteins Involved in the Immune and Inflammatory Responses

The FPL-motif that regulates STAU1^55^ stability is also found in 15 additional proteins. Most of them are linked to the inflammatory response and/or participate in the innate immune response (Table 1 and Figure 7). In addition to its role in cell proliferation, STAU1^55^ was recently linked to inflammation. Through post-transcriptional regulation, STAU1^55^ controls the expression of multiple mRNAs coding for inflammatory and immune response proteins [32,34,52]. For example, STAU1^55^ upregulates the expression of IFIT3, a protein of the MAVS signalosome that interacts with TRAF6 and TBK1 and bridges these proteins to MAVS to enhance innate immunity [53]. Similarly, STAU1^55^ upregulates the translation of netrin-1 mRNA, an inflammation-inducible factor and a proinflammatory protein, in hepatocytes [34]. Consistent with its role in the upregulation of proinflammatory mRNAs, STAU1^55^ expression is itself upregulated in response to lipopolysaccharide-induced embryonic inflammation in mice [49]. These results suggest that STAU1^55^ expression can be induced by inflammatory processes, which leads to signal amplification through the post-transcriptional regulation of STAU1^55^-bound mRNAs. STAU1^55^ expression is upregulated in most cancers compared with normal tissues [23,30]. It was hypothesized that STAU1^55^ upregulation facilitates tumor development via its role in cell cycle phase transitions. This present study suggests that, in addition, STAU1^55^ expression may trigger inflammation and, thus, contribute to tumor growth.

STAU1^55^ has a long tradition of interaction with genomic RNAs and proteins of several RNA viruses. Through these interactions, STAU1^55^ influences viral RNA encapsidation, viral protein association, virus replication, and/or cap-dependent and cap-independent translation of viral proteins [31,54,55,56,57,58]. It also influences the immune and/or antiviral response. For example, following infectious bursal disease virus (IBDV) infection, STAU1^55^ competes with the immune response protein MDA5 for the recognition and binding to viral dsRNAs [31]. As a consequence of its binding to the genomic dsRNA of IBDV, STAU1^55^ prevents MDA5-mediated interferon-β expression, allowing the virus to escape the antiviral response. Similarly, STAU1^55^ competes with the interferon-induced, dsRNA-activated protein kinase PKR for binding dsRNA and, thus, prevents PKR activation [59]. Upon activation by virus infection, PKR blocks viral protein synthesis and activates the NF-κB pathway, thus enhancing the expression of interferon cytokines to spread the antiviral response. In contrast, the binding of STAU1^55^ to endogenous retrovirus dsRNAs induced by the inhibition of DNA methylation stabilizes these dsRNAs. The presence of these dsRNAs in the cells activates proteins of the immune response pathway and the downstream effectors [50]. Therefore, the presence of the FPL-degradation motif in STAU1^55^ would permit us to modulate STAU1^55^-mediated responses during the immune responses. STAU1^55^ degradation may be important to reduce or terminate the inflammation process.

MAP4K1, ABCF1, ADGRG1, GPR83, and OMD are known for their anti-inflammatory functions [36,37,39,40,43]. MAP4K1 is a serine/threonine kinase that negatively regulates the inflammatory and the adaptive immune responses [40]. Upon interaction with the T-cell and B-cell receptors, MAP4K1 triggers a phosphorylation cascade that includes the phosphorylation and then the degradation of the T-cell and B-cell effectors SLP76 and BLNK, respectively. MAP4K1 is also a negative regulator of the antiviral innate immune response by interacting with the ubiquitin ligase DTX4, which, in turn, ubiquitinates TBK1 for degradation [60] (Figure 7). MAP4K1 activity and autophosphorylation then trigger its ubiquitination by the E3 ubiquitin ligase CUL7/fbxw8 and its degradation by the proteasome [41]. Our results now identify TRIM25 as an additional E3 ubiquitin ligase that degrades MAP4K1.

The anti-inflammatory functions of several proteins are often accomplished by regulating the activation of different specialized cells. For instance, ABCF1 induces the transition from M1 macrophages that set a cytokine storm to M2 macrophages that induce the tolerance phase and, thus, switch off inflammation [36]. The transition correlates with the activation of ABCF1 by TRAF6-mediated K63-ubiquitination. ABCF1 possesses an E2 ubiquitin enzyme activity and, via the activation of E3 ligases, activates proteins of the TLR4 signaling pathway, such as SYK and TRAF3 by K63-polyubiquitination, and reduces the inflammatory responses (Figure 7). Similarly, ADGRG1 inhibits the cytotoxicity of the natural killer (NK) cells and production of inflammatory cytokines [37]. Likewise, GPR83 is one of the signature genes expressed in immunosuppressive regulatory T cells (T_reg_), where its expression is regulated by the transcription factor NF-κB [39,61]. GPR83-transduced T cells can interfere with inflammatory responses in vivo, suggesting that GPR83 is involved in T_reg_ induction during immune responses [39].

In contrast, VCAM1, UBE2F, and ABCC11 have proinflammatory roles [35,44,46]. VCAM1 is expressed in response to proinflammatory cytokine stimulation and plays a key role in the immune and inflammatory responses [46]. This transmembrane cell surface adhesion protein controls inflammation-associated leukocyte recruitment and adhesion at inflamed sites and transendothelial migration of leukocytes. Dysregulation of VCAM1 expression is indeed linked to several immunological disorders, such as rheumatoid arthritis and asthma [46]. UBE2F is an E2-NEDD8 ligase involved in the neddylation of proteins. UBE2F activates E3 ubiquitin Cullin-RING ligase 5 (CRL5) by neddylation, triggering K63-ubiquitination of TRAF6 and thus NF-κB activation [62]. Following HIV1 infection, UBE2F is hijacked by the viral protein Vif to promote CRL5-mediated ubiquitination and degradation of the host antiviral proteins APOBEC3. UBE2F, thus, allows the virus to subvert the immune system and achieve a chronic infection [63].

### 3.3. TRIM25 Is the E3 Ubiquitin Ligase That Links a Family of Proteins Involved in Inflammation

TRIM25 is an E3 ubiquitin ligase that can catalyze the addition of K48- and K63-linked polyubiquitin chains for protein degradation or signal transduction, respectively [48,64]. TRIM25 plays several roles in the cells notably during development and differentiation in cancer and in the innate immune system [64,65]. The proximity labeling assays identify TRIM25 as a putative E3 ubiquitin ligase that mediates STAU1^55^ degradation (Table 2). The biochemical characterization of the wild-type and mutated proteins confirms that TRIM25 is involved in STAU1^55^ and MAP4K1 degradation via the FPL-motif (Figure 6). This is consistent with a previous study showing that TRIM25 coimmunoprecipitates with STAU1 [66]. Strikingly, as documented for most of the proteins with the FPL-motif, the main role of TRIM25 is linked to antiviral, immune, and/or inflammatory responses. TRIM25 is part of the RIG-1/interferon pathway [48,64,67] (Figure 7). Upon infection, the pattern recognition receptor RIG-1 recognizes viral RNA molecules and initiates a downstream signaling cascade that leads to interferon production [68]. RIG-1 activity is stimulated by TRIM25-mediated K63-ubiquitination, leading to the activation of the innate immune response [48,67]. TRIM25 also activates the MDA5 pathway via ubiquitination and activation of TRAF6 [69]. Not surprisingly, then, many viruses elaborate specific mechanisms to target/impair the TRIM25/RIG-1 pathway and especially the ubiquitination of RIG-1 by TRIM25 to escape the immune response [70,71,72,73,74,75]. Interestingly, the TurboID assay identifies a cluster of proteins involved in the positive regulation of the RIG-I signaling pathway in the proximity of STAU1^55^ (Appendix A), suggesting that STAU1^55^ is closely related to the immune and/or inflammatory responses. Our results add another layer to the complexity of protein turnover in different cell conditions. They identify TRIM25 as a novel factor in the turnover of the FPL-containing proteins to complement the involvement of other E3 ubiquitin ligases (Table 1) and fine-tune biological processes.

## 4. Materials and Methods

### 4.1. Cell Culture

hTERT-RPE1 (immortalized retinal pigment epithelial cells) and HEK293T (human embryonic kidney cell line) cells were obtained from the American Type Culture Collection. They were cultured in Dulbecco modified Eagle’s medium (DMEM, Wisent Inc, St-Bruno, QC, Canada) supplemented with 10% fetal bovine serum (Wisent), 100 μg/mL streptomycin, and 100 units/mL penicillin (Wisent) under 5% CO_2_ atmosphere.

### 4.2. Plasmids and Cloning Strategies

Plasmid coding for HA-CDH1 was obtained from Dr Michele Pagano [76], and those coding for GFP-Ubi-K11 and GFP-Ubi-K11R from Dr Michel Bouvier [77]. Plasmids coding for Stau1^55^-HA_3_, STAU1^Δ37^-HA_3_, STAU1^Δ46^-HA_3_, STAU1^Δ60^-HA_3_, and STAU1^Δ88^-HA_3_ were previously described [19,20,78,79,80]. RBD2-YFP, RBD2^Δ51^-YFP, RBD2^Δ46^-YFP, and RBD2^Δ37^-YFP were generated by PCR amplification of the N-terminal end of STAU1^55^- HA_3_ using sense and antisense oligonucleotide primers (Appendix A). PCR products were digested with the endonucleases EcoRI and AgeI and inserted into a YFP CMV Topaz vector. For the alanine scanning of amino acids 37 to 51, STAU1^55^-HA_3_ was PCR-amplified using the PfuUltra II polymerase (Agilent Technologies, Toronto, ON, Canada) and oligonucleotide primers (Integrated DNA Technologies, Coralville, IA, USA) (Appendix A). MAP4K1 was cloned in the pcDNA-RSV vector by PCR amplification from pDONR223-MAP4K1 (Appendix A). pDONR223-MAP4K1 was a gift from Drs Hahn and Root (Addgene plasmid #23484; http://n2t.net/addgene:23484 accessed on 30 October 2019; RRID: Addgene_23484) [81]. The PCR product was cloned at the XhoI and NotI sites. An HA_3_ tag was then inserted at the Not1 site as described [20]. The MAP4K1^F651A^-HA_3_ mutant was generated using the PfuUltra II polymerase and specific oligonucleotide primes (Appendix A).

To produce STAU1^55^-HA_3_-biotin ligase (BioID2), STAU1^55^-HA_3_ in pCDNA3 RSV was PCR-amplified using oligonucleotide primers (IDT, Coralville, IA, USA) (Appendix A). The PCR product was digested with the restriction enzymes NheI (Thermo Scientific, St-Laurent, QC, Canada) and SacI (NEB, Whitby, ON, Canada) and ligated into MCS-BioID2-HA (a gift from Dr Kyle Roux, Addgene plasmid #74224; http://n2t.net/addgene:74224 accessed on 6 June 2018; RRID: Addgene_74224) [82]. STAU1^55^-BioID2 was then pulled off the vector by digestion with NheI and AgeI and inserted into pMSCVpuro. To produce STAU1^55^-HA_3_-TurboID, STAU1^55^-HA_3_ in pCDNA3 RSV was PCR-amplified using oligonucleotide primers (IDT, Coralville, IA, USA) (Appendix A). The PCR product was digested with the restriction enzymes NotI (NEB, Whitby, ON, Canada) and NheI (Thermo Scientific, St-Laurent, QC, Canada) and ligated into V5-TurboID-NES_pCDNA3 (a gift from Dr Alice Ting, Addgene plasmid #107169; http://n2t.net/addgene:107169 accessed on 13 April 2019; RRID: Addgene_107169) [83] with T4 ligase (NEB, Whitby, ON, Canada). STAU1^55^-TurboID was then extracted by digestion with NotI (NEB, Whitby, ON, Canada) and BglII, treated with PNK (Thermo Scientific, St-Laurent, QC, Canada), and inserted in pMSCVpuro after digestion with HpaI (Thermo Scientific, St-Laurent, QC, Canada). Positive clones were selected and digested with NotI to insert the HA_3_ tag as described [20]. siRNAs against the ubiquitin ligases TRIM25, TRIP12, and ZNF598 were purchased from IDT. The oligonucleotide primers used to quantify their expression are listed in Appendix A.

### 4.3. Antibodies and Reagents

Antibodies against GFP (11814460001) were purchased from Roche (Oakville, ON, Canada) and used to detect YFP-tagged proteins since the two proteins are identical except for one amino acid and anti-GFP antibody perfectly recognizes YFP. Anti-STAU1 was previously described [84]. Anti-β-actin (A5441) and anti-α-tubulin (T5168) were obtained from Sigma-Aldrich (Aokville, ON, Canada). Anti-HA (SC7392) and anti-CDC20 (SC13162) antibodies were purchased from Santa Cruz Biotechnology (Dallas, TX, USA). All primary antibodies were used at 1:1000 dilution. MG132 (C2211) and DMSO were purchased from Millipore-Sigma (Aokville, ON, Canada).

### 4.4. Western Blot Analysis

Total-cell extracts were prepared in Laemli lysis buffer (25 mM Tris-Cl pH 7.4, 1% SDS), and protein concentrations were determined using the Pierce^TM^ BCA protein assay (23223) (Thermo Scientific, St-Laurent, QC, Canada). Cell extracts (10–20 μg) were analyzed by Western blotting. Data were collected either on X-ray films (Fujifilm—Christie Innomed, St-Eustache, QC, Canada) or with the ChemiDoc MP Imaging System (Bio-Rad Laboratories—St-Laurent, QC, Canada), and Western blot signals were quantified with the ImageJ or ImageLab (Bio-Rad Laboratories—St-Laurent, QC, Canada) software (version number: 1.53c), respectively.

### 4.5. Stabilization, Degradation, and Ubiquitination Assays

For the analysis of STAU1^55^ stabilization, HEK293T cells were transfected with STAU1^55^ or its deletion mutants and treated with the proteasome inhibitor MG132 (20 μM) for 8 h. For degradation assay, HEK293T cells were cotransfected with plasmids coding for STAU1^55^-HA_3_, RBD2-YFP, or their mutants and plasmids coding for the empty vector, CDC20-HA or CDH1-HA. For ubiquitination assays, HEK293T cells were cotransfected with plasmids coding for STAU1^55^ or its mutants and for GFP-UBI or GFP-UBI-K11R. Twenty-four hours after transfection, cells were treated with MG132 as described above. Protein extracts were prepared in the lysis buffer, cleared by sonication for 20 s, and centrifuged at 15,000× *g* for 15 min. Proteins were analyzed by Western blotting.

### 4.6. RNA Isolation and RT-qPCR

RNA isolation was performed by homogenization of HEK293T cells with TRIZOL reagent (Ambion, St-Laurent, QC, Canada). After precipitation, purified RNAs were resuspended in 40–50 μL of purified water (RNA free) and digested with DNAse using the TURBO DNA-free kit (Ambion, St-Laurent, QC, Canada). An amount of 1 μg of the resulting RNA was used for the reverse transcription reaction using the RevertAid H Minus First Strand cDNA Synthesis Kit (Thermo Scientific, St-Laurent, QC, Canada) and oligo(dT). qPCR was performed in triplicates using the Luna^®^ Universal qPCR Master Mix (NEB—Whitby, ON, Canada) in a 96-well plate and run on a LightCycler 96 (Roche—Oakville, ON, Canada). Normalization was performed using the average of HPRT and RPL22 gene expression.

### 4.7. BioID2/TurboID Immunoprecipitation

hTERT-RPE1 cells were used to generate a stable cell line expressing Stau1^55^-HA_3_-BioID2 or STAU1^55^-HA_3_-TurboID by retroviral infection. After puromycin selection, cells were incubated in the presence of 50 μM biotin (SIGMA—Oakville, ON, Canada) for either 16 (BioID2) or 3 h (TurboID). For the TurboID assay, cells were first synchronized either in mitosis with nocodazole or in G_1_/S by a double thymidine block as described [19] before incubation with biotin. Biotinylated proteins were pulled down with 250 μL of prewashed streptavidin-coated magnetic beads (Dynabeads MyOne Streptavidin T1). Beads were then washed six times with RIPA lysis buffer (25 mM Tris-HCl, pH 7.6, 150 mM NaCl, 1% NP-40, 1% sodium deoxycholate, 0.1% SDS, 1% Triton X-100, 1× protease inhibitor cocktail, 1× PhosStop), once with 1 M KCl, once with 8 M urea in 10 mM Tris-HCl (pH 8.0), thrice with modified RIPA lysis buffer (50 mM Tris pH 8, 150 mM NaCl, 0.02% SDS, 0.5% sodium deoxycholate, and 0.2% Triton X-100), and once with 0.1 M Na_2_CO_3_. Beads were then resuspended in 100 μL of 50 mM ammonium bicarbonate solution.

### 4.8. Protein Digestion and LC-MS/MS

Proteins on beads were digested overnight with 1 μg of sequencing grade modified trypsin (Promega—Madison, WI, USA). Resulting peptides were reduced in 9 mM dithiothreitol for 30 min at 37 °C and alkylated with 17 mM iodoacetamide for 20 min at room temperature. Peptides were then desalted and eluted in 10% ammonium hydroxide/90% methanol and resuspended in 5% ferulic acid. Peptides were injected in the Easy-nLC II system (Proxeon Biosystems—St-Laurent, QC, Canada) for chromatography using two buffers, 0.2% formic acid and 0.2% formic acid plus 90% acetonitrile. Peptides were sent to a Q Exactive mass spectrometer (Thermo Scientific, St-Laurent, QC, Canada) through a Nanospray Flex Ion Source set to 1.3–1.8 kV for the nanospray and to 50 V for the S-lens at a capillary temperature of 225 °C. The MS survey spectra was acquired in the Orbitrap with a resolution of 70,000. The most intense peptide ions were fragmented, and the MS/MS was also analyzed in the Orbitrap. The mass spectrometry proteomics data have been deposited in the ProteomeXchange Consortium via the PRIDE [85] partner repository with the dataset identifiers PXD036675 and 10.6019/PXD036675.

## 5. Conclusions

The fine-tuning of biological processes needs the coordinated regulation of effector proteins by upstream modulators. We show that the E3 ubiquitin ligase TRIM25 can be one of these modulators in charge of a family of proteins involved in inflammation. Interestingly, although the number of proteins in the family is relatively modest, the inflammatory signal initiated by TRIM25 can be highly amplified by the FPL-motif-containing proteins. In fact, these proteins control post-transcriptional and post-translational processes and/or trigger signal transduction (RNA-binding protein, kinases, transporters, receptors, ubiquitin ligase) (Table 1). It will be interesting to define the type of ubiquitination chains (K48; K63) on proteins of the family as a clue to whether the ubiquitination process is used to activate or attenuate the inflammatory process. The relevance of TRIM25 in the regulation of the family is strengthened by the observation of a direct interaction with STAU1^55^ [66]. Whether TRIM25 interacts with other proteins of the family via the FPL-motif is still an open question. Similarly, it will be relevant to determine how TRIM25 affects the pathways normally controlled by the proteins and whether this family of proteins is involved in cancer-related inflammation. Once the role of the FPL-motif is better understood, it will be essential to determine whether it can serve as platforms for drug targeting against cancer and/or inflammatory diseases.

## Figures and Tables

**Figure 1 ijms-23-11588-f001:**
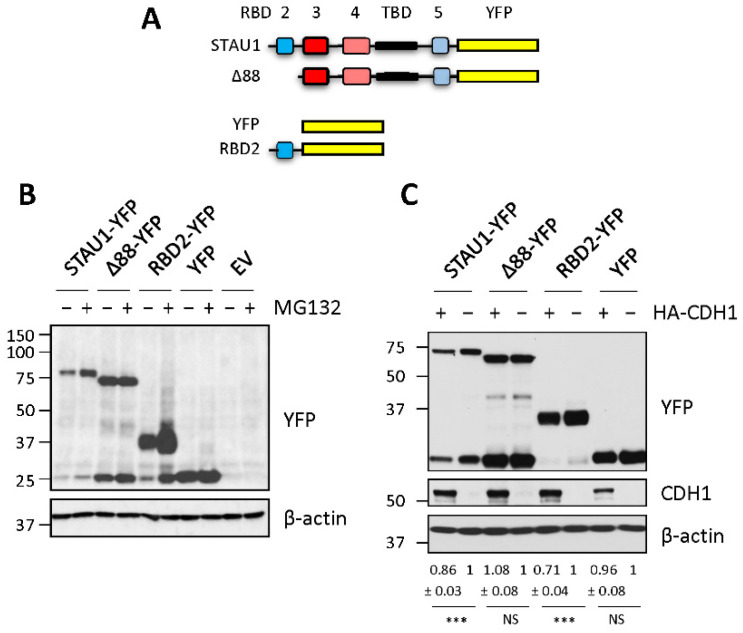
RBD2 is sufficient to induce degradation of STAU1^55^ by APC/C-UPS. (**A**) Schematic representation of STAU1^55^-YFP and the deletion mutants. STAU1, STAU1^55^; Δ88, STAU1^55^ lacking the first N-terminal 88 amino acids; RBD2, RNA-binding domain 2 corresponding to the first N-terminal 88 amino acids of STAU1^55^; YFP, yellow fluorescent protein. Red boxes, RNA-binding domains (RBD); blue boxes, regions with RNA-binding consensus sequence but lacking RNA-binding activity in vitro; yellow boxes, YFP; black boxes, tubulin-binding domain (TBD). (**B**) HEK293T cells expressing STAU1^55^-YFP, deletion mutants, YFP, or an empty vector (EV) were grown in the presence (+) or absence (−) of the proteasome inhibitor MG132 for 8 h. Protein extracts were analyzed by Western blotting. (**C**) HEK293T cells were cotransfected with plasmids coding for HA-CDH1 (+) or an empty vector (−) and plasmids coding for STAU1^55^-YFP, STAU1^Δ88^-YFP (Δ88-YFP), RBD2-YFP, or YFP. Degradation of the proteins in the presence (+) or absence (−) of HA-CDH1 was observed by Western blotting. Western blots using anti-CDH1 antibody confirm the expression of HA-CDH1. The blots are representative of three independently performed experiments that gave similar results. The ratio of the amounts of the expressed proteins to that of actin in the absence or presence of HA-CDH1 was calculated, and the ratio in the absence of HA-CDH1 was arbitrarily fixed to 1. The numbers represent the means ± standard deviation of the relative amounts of the proteins in three independently performed experiments. ***, *p*-value ≤ 0.001 (Student *t*-test). NS, not significant.

**Figure 2 ijms-23-11588-f002:**
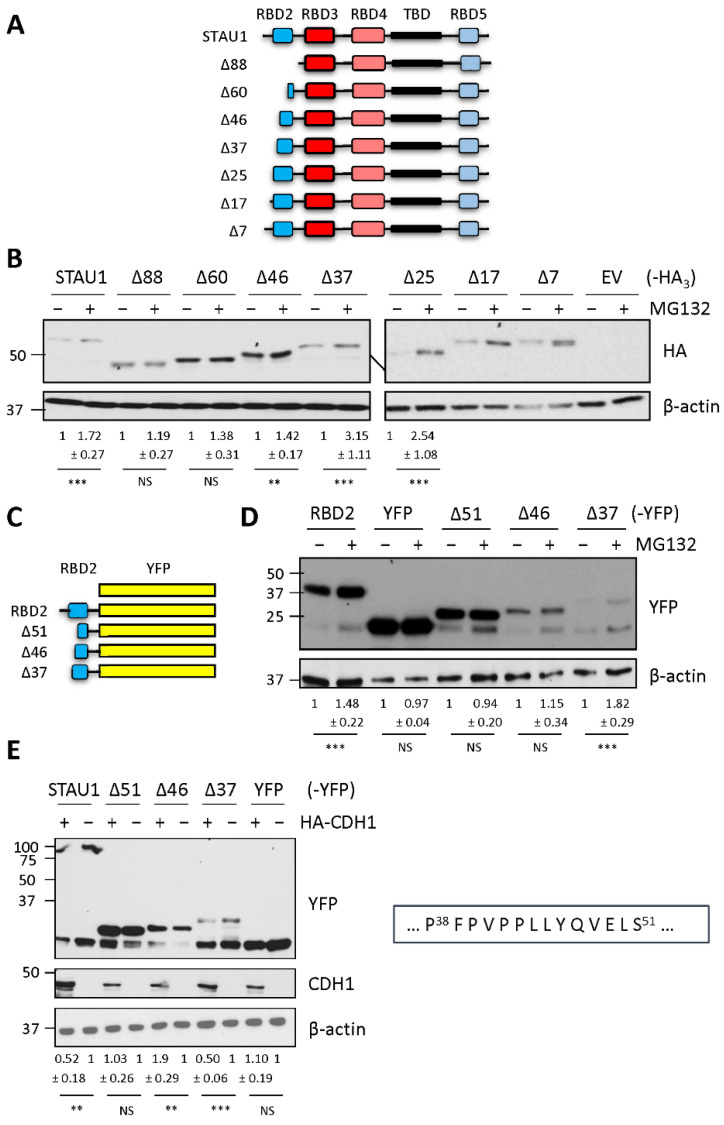
The degradation motif is located within amino acids 38 and 51. (**A**) Schematic representation of STAU1^55^ and mutants with progressive deletions of the first N-terminal 88 amino acids (see legend in Figure 1). (**B**) Plasmids coding for STAU1^55^-HA_3_ or deletion mutants were transfected in HEK293T cells. After 24 h, cells were treated with 20 μM MG132 for 8 h (+). Protein extracts were analyzed by Western blotting using anti-HA antibody. (**C**) Schematic representation of RBD2 and the deletion mutants fused to YFP. (**D**) HEK293T cells were transfected with plasmids coding for RBD2-YFP or the deletion mutants and treated, 24 h later, with 20 μM MG132 for 8 h (+). Protein extracts were analyzed by Western blotting using. (**E**) HEK293T cells were cotransfected with plasmids coding for HA-CDH1 or an empty vector and plasmids coding for STAU1^55^-YFP, RBD2-YFP deletion mutants, or YFP. Degradation of the proteins in the presence (+) or absence (−) of HA-CDH1 was observed by Western blotting. Western blots using anti-HA antibody confirm the expression of HA-CDH1. The box on the right shows the minimal amino acid sequence responsible for STAU1^55^ degradation. The numbers below the gels represent the means ± standard deviation of the relative amounts of the proteins in three independently performed experiments. The ratio of the amount of the proteins to that of actin in the absence (−) of MG132 (**B**,**D**) or HA-CDH1 (**E**) was arbitrarily fixed to 1. ***, *p*-value ≤ 0.001. **, *p*-value ≤ 0.01 (Student *t*-test). NS, not significant.

**Figure 3 ijms-23-11588-f003:**
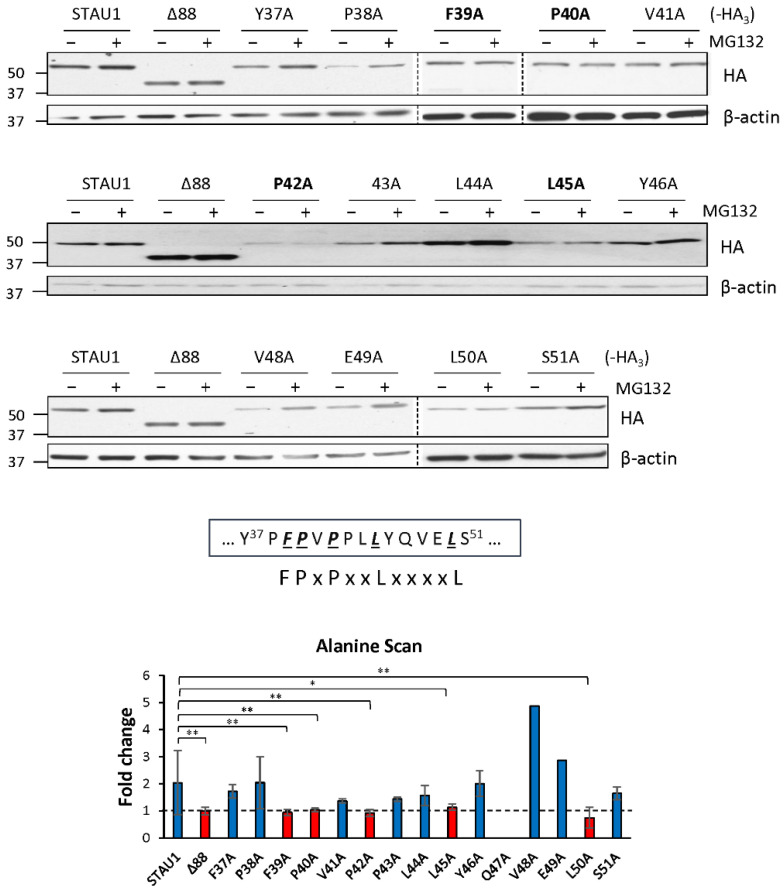
Identification of the STAU1^55^ degradation motif. HEK293T cells were transfected with plasmids coding for STAU1^55^-HA_3_ or mutants in the putative degradation motif and treated, 24 h later, with 20 μM MG132 for 8 h (+). Protein extracts were analyzed by Western blotting using anti-HA antibody. The graph represents the means ± standard deviation of the relative amounts of the proteins in three independently performed experiments. The ratio of the amount of STAU1^55^ and STAU1^55^ mutants to that of actin in the absence (−) of MG132 was arbitrarily fixed to 1. **, *p*-value ≤ 0.01. *, *p*-value ≤ 0.05 (Student *t*-test). The box below the gels highlights the amino acids required for STAU1^55^ degradation by UPS.

**Figure 4 ijms-23-11588-f004:**
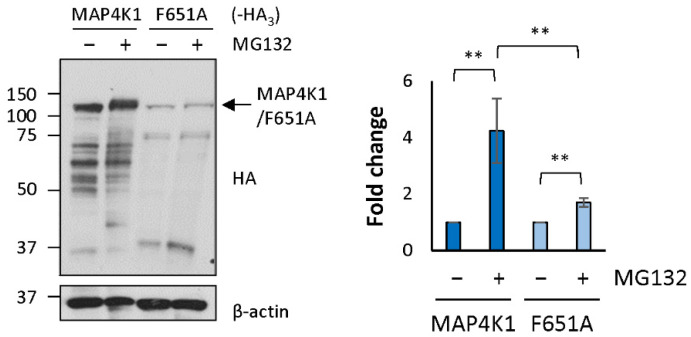
MAP4K1 is degraded via the conserved degradation motif. HEK293T cells were transfected with plasmids coding for MAP4K1-HA_3_ or MAP4K1^F651A^-HA_3_ (F651A) and, 24 h later, treated with the proteasome inhibitor MG132 for 8 h (+). Protein extracts were analyzed by Western blotting using anti-HA antibody. The graph on the right represents the means ± standard deviation of the relative amounts of the proteins in three independently performed experiments. The ratio of the amount of MAP4K1-HA_3_ and MAP4K1^F651A^-HA_3_ to that of actin in the absence (−) of MG132 was fixed to 1. **, *p*-value ≤ 0.01 (Student *t*-test).

**Figure 5 ijms-23-11588-f005:**
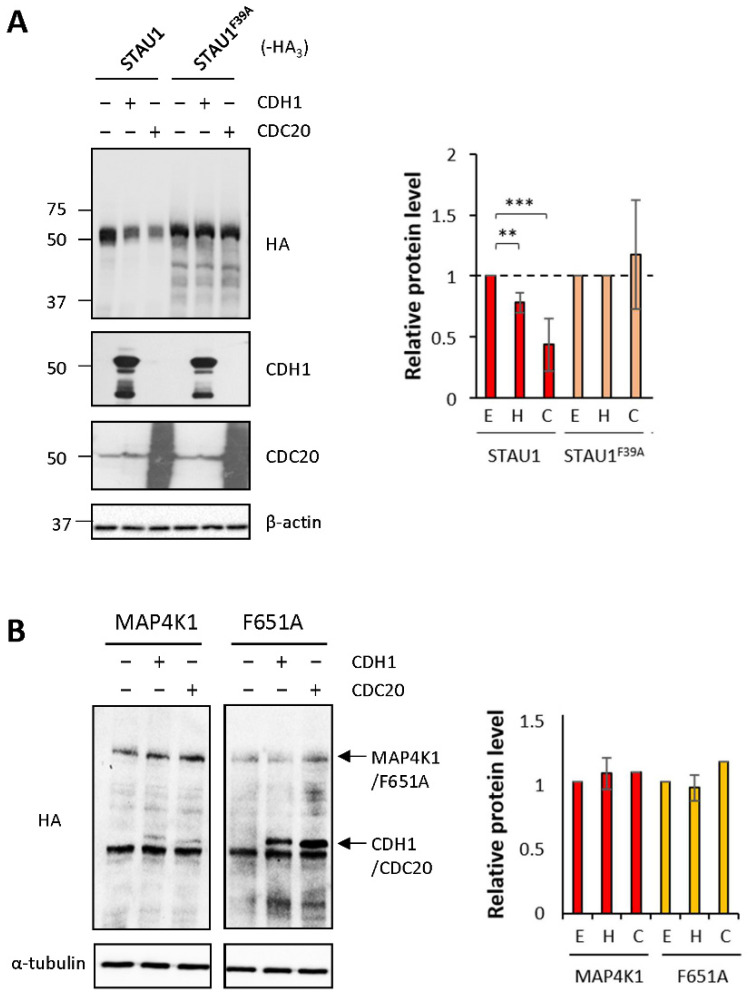
CDC20 and CDH1 recognize the FPL-motif of STAU1^55^ but not of MAP4K1. HEK293T cells were cotransfected with plasmids coding for STAU1^55^-HA_3_, STAU1^F39A^-HA_3_ (**A**)**,** MAP4K1-HA_3_, or MAP4K1^F651A^-HA_3_ (**B**) and plasmids coding for CDC20 or CDH1. Protein extracts were analyzed by Western blotting using anti-HA antibody. The graphs on the right represent the means ± standard deviation of the relative amounts of the proteins in three independently performed experiments. The ratio of the amount of the proteins to that of actin (**A**) or tubulin (**B**) in the absence of cotransfected proteins was fixed to 1. ***, *p*-value ≤ 0.001. **, *p*-value ≤ 0.01 (Student *t*-test). NS, not significant. E, empty vector; H, CDH1; C, CDC20.

**Figure 6 ijms-23-11588-f006:**
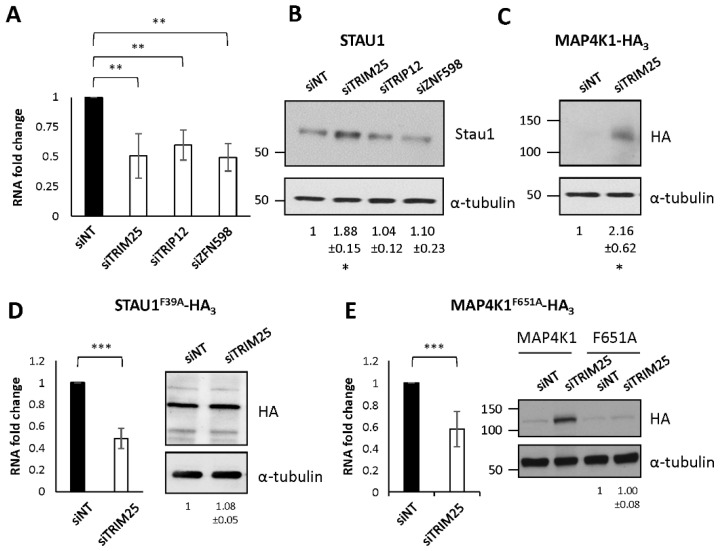
TRIM25 is involved in STAU1^55^ and MAP4K1 degradation via the FPL-motif. (**A**) HEK293T cells were transfected with siRNA against the E3 ubiquitin ligases TRIM25, TRIP12, or ZNF598. After 24 h, RNAs were purified from cell extracts and the amounts of mRNA coding for the E3 ligases quantified by RT-qPCR. Their amounts were normalized over that of HPRT and RPL22. The ratio in cells transfected with nontargeting siRNAs (siNT) was arbitrarily fixed to 1. **, *p* value ≤ 0.01 (Student *t*-test). (**B**) siRNA-transfected cells were lysed, and the amounts of endogenous STAU1^55^ proteins analyzed by Western blot using anti-STAU1 and anti-tubulin antibodies. (**C**) HEK293T cells were cotransfected with plasmid coding for MAP4K1-HA_3_ and siRNA against TRIM25. Protein extracts were analyzed by Western blotting using anti-HA and anti-tubulin antibodies. The ratio of the amounts of the expressed proteins on that of tubulin in control cells was arbitrary fixed to 1. The numbers represent the means ± standard deviation of the relative amounts of the proteins in three independently performed experiments. *, *p* value ≤ 0.05 (Student *t*-test). (**D**,**E**) HEK293T cells were transfected with siRNA against the E3 ubiquitin ligase TRIM25 and plasmids coding for STAU1^F39A^-HA_3_ (**D**) or MAP4K1^F651A^-HA_3_ (**E**). MAP4K1-HA_3_ was used as control. After 24 h, RNAs were purified from cell extracts and TRIM25 mRNA quantified by RT-qPCR (left panels). The amount was normalized as in (**A**). Protein levels were analyzed by Western blot (right panels) using anti-HA and anti-tubulin antibodies. The blots are representative of three independently performed experiments that gave similar results. The numbers below the gels represent the means ± standard deviation of the relative amounts of the proteins in three independently performed experiments. The ratio of the amount of the proteins to that of tubulin in siNT-transfected cells was arbitrary fixed to 1. ***, *p* value ≤ 0.001 (Student *t*-test).

**Figure 7 ijms-23-11588-f007:**
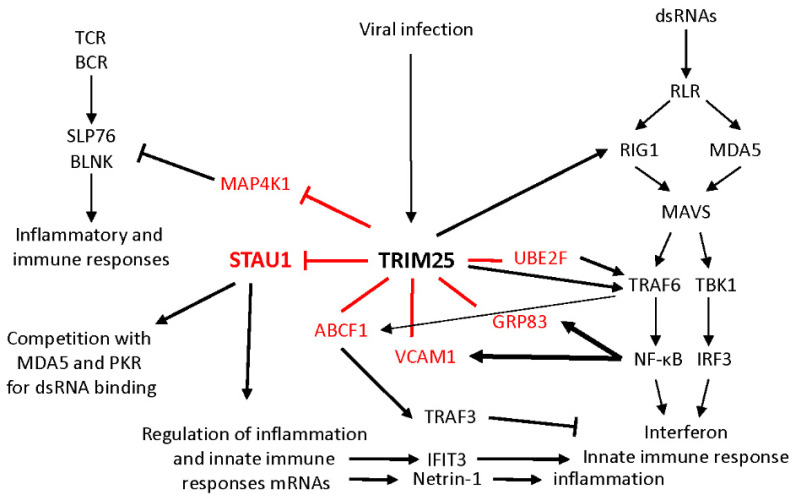
A family of proteins linked by the FPL-motif is involved in inflammation. Schematic representation of the interactions between proteins of the FPL-motif family, the E3 ubiquitin ligase TRIM25, and proteins of the adaptive (left pathway) and innate (right pathway) immune response pathways. TRIM25 activates the RIG1 pathway of the innate immune response via K63-ubiquitination of RIG1 and TRAF6. On the other hand, TRIM25 causes the degradation of MAP4K1, an inhibitor of the adaptive immune response via the T-cell (TCR) and B-cell (BCR) receptor pathways. It also degrades STAU1^55^. Through competition for dsRNA binding, STAU1^55^ prevents the activation of proteins of the innate immune response, such as MDA5 and PKR. STAU1^55^ also up- or downregulates the expression of multiple mRNAs coding for proteins of the inflammatory and immune response pathways. For example, the STAU1^55^-mediated upregulation of IFIT3 and netrin-1 leads to enhanced immune response via the activation of the MAVS pathway and inflammation, respectively. TRIM25 also targets other proteins (in red). Whether it activates (K63-ubi) or degrades (K48-ubi) these proteins is still unresolved. UBE2F activates the NF-κB pathway and interferon production by K63-ubiquitination of TRAF6. ABCF1 enhances anti-inflammatory response by K63-ubiquitination-mediated activation of TRAF3. In turn, NF-κB binds the promotor of the *VCAM1* and *GRP83* genes and upregulates their expression.

**Table 1 ijms-23-11588-t001:** List of proteins with the F-P-x-P-x(2)-L-x(4)-L motif.

Gene Symbol	Gene Title	Links to Inflammation	Inflammatory Process	E3 Ubi Ligases	Number of Ubi Sites ^a^	SSV ^b^
ABCC11	ATP-binding cassette subfamily C member 11	[35]	Pro-			
ABCF1	ATP-binding cassette subfamily F member 1	[36]	Anti-	TRAF6, cIAP1/2 [36]	36	2.04
ADGRG1	Adhesion G protein-coupled receptor G1	[37]	Anti-		2	
CTAGE4	CTAGE family, member 4	[38]			1	
CTAGE8	CTAGE family, member 8				1	
CTAGE9	CTAGE family, member 9				1	
GPR83	G protein-coupled receptor 83	[39]	Anti-			
KBTBD13	Kelch repeat and BTB domain-containing protein 13				1	
LINGO4	Leucine-rich repeat and immunoglobulin-like domain-containing Nogo receptor-interacting protein 4					
MAP4K1	Mitogen-activated protein kinase kinase kinase kinase 1	[40]	Anti-	CUL7/Fbxw8 [41]	7	1.74
MCHR2	Melanin-concentrating hormone receptor 2	[42]	Pro-			
OMD	Osteomodulin	[43]	Anti-			1.76
PRRC1	Proline-rich coiled-coil 1				3	1.129
STAU1	Staufen double-stranded RNA-binding protein 1	[32]	Pro-/anti-	APC/C [19]	14	
UBE2F	Ubiquitin-conjugating enzyme E2F	[44]	Pro-	CUL3, PARK2 [45]	6	
VCAM1	Vascular cell adhesion molecule 1	[46]	Pro-		2	2.56

^a^ Number of ubiquitination sites as reported (BioGRID), https://thebiogrid.org/ (accessed on 17 August 2022); ^b^ SSV: score standardized value (genes associated with inflammation) [47], https://maayanlab.cloud/Harmonizome/gene_set/Inflammation/CTD+Gene-Disease+Associations.

**Table 2 ijms-23-11588-t002:** List of E3 ubiquitin ligases in the proximity of STAU1^55^. Total spectrum count in different conditions.

	BioID2		TurboID (G1/S)		TurboID (Mitosis)	
	Ctrl	STAU1^55^	Ctrl	STAU1^55^	Ctrl	STAU1^55^
Cul4B	2	3	13	4		
RBBP6	0	2			44	85
TRIP12	0	1	0	6	11	36
TRIM25	0	8	13	32	4	65
TRIM56	0	1	0	2		
SMURF2			0	1	0	3
ZNF598	0	1	0	5	6	55
CDC20					3	8
CDH1						

## Data Availability

The data presented in this study are available in the manuscript and Appendix A. The mass spectrometry proteomics data have been deposited in the ProteomeXchange Consortium via the PRIDE [85] partner repository with the dataset identifiers PXD036675 and 10.6019/PXD036675.

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
