# Peer review of "A Degradation Motif in STAU1 Defines a Novel Family of Proteins Involved in Inflammation"

_ijms, 2022, doi:10.3390/ijms231911588_

Round 1
Reviewer 1 Report
Quesada’s paper identifies a novel motif, FPL-motif, responsible for ubiquitin-mediated degradation. From what has been found that STAU1 is degraded by APC/C complex, this paper further finds the short fragment AA38-50 is essential for STAU1 degradation. In addition, through mutation assay, the important residues are defined and FPxPxxLxxxxL consensus sequence is suggested. Interestingly, 15 more proteins are found to have this motif and MAP4K1 is verified. It is an interesting finding to define a small protein family containing FPL motif. However, comments are raised for fastening the conclusion.
1. The paper provides biochemical data for the protein degradation. Without cell experiments, inflammation is title and so much discussion (3.2, 3.3, 3.4) in discussion section are not suitable. Too many references are included.
2. STAU1 contains FPL-motif and is degraded by APC/C and TRIM25. MAP4K1 also contains FPL-motif and is degraded by TRIM25, but not APC/C. This difference should be explained. Moreover, total 16 proteins contain FPL-motif. Are APC/C, TRIM25, or other E3 ligases responsible for their degradation? Answer these questions will enhance the importance of the FPL-motif.
3. Total 8 Lys residues are in RBD2 domain. Mutation with deletion 37 residues in N-terminal (delta 37) of RBD2 is degraded by APC, suggesting Lys 2, 5, 6, 10 are not involved. The left 4 Lys are Lys60, 62, 68, and 74 are candidates for ubiquitination modification. It is interesting to know whether the distance between FPL-motif and Lys residues is essential for the regulation.
4. Fig 1C is meaningless and co-IP experiments are required.
5. Fig 1D should include overexpressing of CDC20 and APC/C.
6. Fig 2E should include overexpressing of CDC20 and APC/C. And importantly, the interaction resulting should be provided.
7. Table 1 should provide the information similar to supplemental Fig 2B and C for all proteins.
8. Fig 4A, mutation causes less MAP4K1 F651A expressed in HEK293T cells. Other important residues in FPL-motif should be selected for mutation assay.
Author Response
We thank the reviewers for their positive and constructive comments on the paper.
Reviewer 1:
Comments and Suggestions for Authors
Quesada’s paper identifies a novel motif, FPL-motif, responsible for ubiquitin-mediated degradation. From what has been found that STAU1 is degraded by APC/C complex, this paper further finds the short fragment AA38-50 is essential for STAU1 degradation. In addition, through mutation assay, the important residues are defined and FPxPxxLxxxxL consensus sequence is suggested. Interestingly, 15 more proteins are found to have this motif and MAP4K1 is verified. It is an interesting finding to define a small protein family containing FPL motif. However, comments are raised for fastening the conclusion.
- The paper provides biochemical data for the protein degradation. Without cell experiments, inflammation is title and so much discussion (3.2, 3, 3.4) in discussion section are not suitable. Too many references are included.
The discussion was shortened and the number of reference reduced. However, we think that the term inflammation can stay in the title. The first description of most of these proteins refers to inflammation.
- STAU1 contains FPL-motif and is degraded by APC/C and MAP4K1 also contains FPL-motif and is degraded by TRIM25, but not APC/C. This difference should be explained. Moreover, total 16 proteins contain FPL-motif. Are APC/C, TRIM25, or other E3 ligases responsible for their degradation? Answer these questions will enhance the importance of the FPL-motif.
We believe that MAP4K1 is not degraded by APC/C because their subcellular localization is not the same. APC/C is a nuclear protein and is activated during mitosis. MAP4K1 is located in membranes and no report links this protein to mitosis. We added a sentence in the discussion.
Characterization of all other 14 proteins for their mechanisms of degradation is indeed the next step to follow. However, we believe that it represents a long-term project outside the scope of this paper. We added a brief discussion on the complexity of protein homeostasis via binding of multiple E3 ubiquitin ligases.
- Total 8 Lys residues are in RBD2 Mutation with deletion 37 residues in N- terminal (delta 37) of RBD2 is degraded by APC, suggesting Lys 2, 5, 6, 10 are not involved. The left 4 Lys are Lys60, 62, 68, and 74 are candidates for ubiquitination modification. It is interesting to know whether the distance between FPL-motif and Lys residues is essential for the regulation.
Proteomic analysis revealed that lys74 is ubiquitinated (24 nucleotides downstream of the FPL motif). Mutagenesis assays will be required to determine if this residue (or any other ones) is necessary and sufficient and how it is involved in STAU1 degradation.
Interestingly, a lysine residue is located 22 nucleotides downstream of the FPL motif in MAP4K1. As mentioned by this reviewer, it will be interesting to do mutagenesis and structural studies to determine if the distance between the FPL-motif and the Lys residue is involved in the regulation.
- Fig 1C is meaningless and co-IP experiments are
We agree. We removed fig 1C.
- Fig 1D should include overexpressing of CDC20 and APC/C.
Overexpression of APC/C is a difficult experiment since APC/C is a complex E3 ligase composed of 13 subunits. Its activation is also highly regulated by different members of the CDC20 family, pseudosubstrate inhibitors, protein kinases and phosphatases and the spindle assembly checkpoint.
- Fig 2E should include overexpressing of CDC20 and APC/C. And importantly, the interaction resulting should be
Overexpression of APC/C is a difficult experiment since APC/C is a complex E3 ligase composed of 13 subunits. Its activation is also highly regulated by different members of the CDC20 family, pseudosubstrate inhibitors, protein kinases and phosphatases and the spindle assembly checkpoint.
- Table 1 should provide the information similar to supplemental Fig 2B and C for all
This information was added in supplemental figures for all proteins (Supp Fig 2 D-M).
- Fig 4A, mutation causes less MAP4K1 F651A expressed in HEK293T Other important residues in FPL-motif should be selected for mutation assay.
It is not clear if the reduced expression of the mutant is a consequence of the mutation or the transfection assay. Anyway, we always managed to express low amounts of the proteins to avoid protein saturation and to optimize detection of even small variation in protein amounts when cell culture conditions are changing.
Reviewer 2 Report
In this study, Quesada and DesGroseillers have revealed a novel FPL-motif in STAU1 serving as ubiquitylation targeting motif for the degradation of STAU1. The FPL-motif is particularly found in inflammation/immune response related proteins; one of them MAP4K1, was indeed degraded via FPL-motif dependent ubiquitylation. Moreover, proximity-biotinylation based interactome analysis revealed further potential E3 ubiquitin ligases that might involve in the turnover of STAU1, as well as FPL-motif including proteins. One of the qualified hits, TRIM25 was found to ubiquitylate and trigger the degradation of STAU1 and MAP4K1 but not their FPL-motif mutated counterparts. All in all, it is a mechanistically well-investigated study. I do have a couple of major, but mostly minor questions and suggestions.
Major points:
Polyubiquitylation assay of RBD2-YFP (Fig.1C) does not clearly indicate whether the smear belongs to RBD2. For that, authors should perform HA tag immunoprecipitation followed by a western immunoblotting using YFP/GFP antibody. Also, the conditions should be as same as used in Fig.1B, to be able to clearly conclude on the specific polyubiquitylation of RBD2.
It would have been better to have FPL-motif deleted STAU1 for the proximity-biotinylation based interactome analysis to distinguish the FPL-targeting E3 ubiquitin ligases.
Minor points:
The description of the degradation of ubiquitylated proteins (lines 37-38) having some shortfalls: Not always polyubiquitylation is required for the degradation; small proteins are mostly getting degraded by monoubiquitylation. Also, 26S proteasome is encompassing only a small fraction (15-25%) of whole proteasome family in the eukaryotic cells. The rest is covered by 20S and 30S. Authors are kindly asked to extend the description a bit detailed and more inclusive.
General question: As authors mentioned in the intro that CDH1 helps APC/C to target proteins; but how CDH1 overexpression helps to increase the turnover rate of STAU1 and RBD2 domain while the protein level of APC/C complex remains unchanged? Do APC/C protein levels get changed upon CDH1 overexpression and/or upon different STAU1 wt and deletion constructs’ overexpression (Fig. 1D and 2E)? Have authors considered to include CDH1 knockdown condition besides to CDH1 overexpression?
Authors have mentioned only two sentences (Lines 61-64) about STAU1 (63kDa); then directly switched to isoform-2 STAU1 (55kDa) for the rest of the manuscript. When compared these two sequences, STAU1 (63kDa) is also containing the FPL-motif: What are the functional differences between these two variants? What is the scientific reasoning to particularly investigate the STAU1 (55kDa) but not both isoforms in the context of FPL-motif dependent turnover?
The “F37A” typo should be corrected as “Y37A” in the upper blot of Fig.3.
Authors are kindly asked to submit the raw mass spectrometry files to a public depository such as MassIVE or PRIDE.
What exactly makes the STAU1 but not MAP4K1 targeted by APC/C while both having the FPL-motif? What residues could be the determinant in the FPL-motif (or at some upstream/downstream residues of FPL-motif) to make it highly-targeted to certain substrates? Or could that be subcellular restriction of the MAP4K1 localization that does not permit to be targeted by APC/C?
A suggestion for the discussion: A recent study revealed that C-terminal glutamine residues could get targeted by TRIM family protein TRIM7 (PMID: 35982226). When the C-terminal glutamine residues (477-QQ-478) of STAU1 considered in that context, it would be nice to discuss the potential involvement of other E3 ligases, perhaps other TRIM family proteins in the turnover of STAU1.
Author Response
We thank the reviewers for their positive and constructive comments on the paper.
Reviewer 2:
Comments and Suggestions for Authors
In this study, Quesada and DesGroseillers have revealed a novel FPL-motif in STAU1 serving as ubiquitylation targeting motif for the degradation of STAU1. The FPL-motif is particularly found in inflammation/immune response related proteins; one of them MAP4K1, was indeed degraded via FPL-motif dependent ubiquitylation. Moreover, proximity-biotinylation based interactome analysis revealed further potential E3 ubiquitin ligases that might involve in the turnover of STAU1, as well as FPL-motif including proteins. One of the qualified hits, TRIM25 was found to ubiquitylate and trigger the degradation of STAU1 and MAP4K1 but not their FPL-motif mutated counterparts. All in all, it is a mechanistically well-investigated study. I do have a couple of major, but mostly minor questions and suggestions.
Major points:
Polyubiquitylation assay of RBD2-YFP (Fig.1C) does not clearly indicate whether the smear belongs to RBD2. For that, authors should perform HA tag immunoprecipitation followed by a western immunoblotting using YFP/GFP antibody. Also, the conditions should be as same as used in Fig.1B, to be able to clearly conclude on the specific polyubiquitylation of RBD2.
We agree. We removed fig 1C.
It would have been better to have FPL-motif deleted STAU1 for the proximity- biotinylation based interactome analysis to distinguish the FPL-targeting E3 ubiquitin ligases.
We agree that many negative controls could have been used. However, none of them is ideal and therefore it is difficult to predict which one will be the best to use. In contrast to IP which identifies direct interaction between proteins (thus mutation of the interacting domain will prevent detection), proximity assays will identify proteins in proximity even those that do not directly interact. Therefore, deletion of the FPL-motif may not necessarily be informative in this assay (it may even be misleading if the two proteins stay longer in proximity looking for an interaction that can not occur due to the mutation).
Minor points:
The description of the degradation of ubiquitylated proteins (lines 37-38) having some shortfalls: Not always polyubiquitylation is required for the degradation; small proteins are mostly getting degraded by monoubiquitylation. Also, 26S proteasome is encompassing only a small fraction (15-25%) of whole proteasome family in the
eukaryotic cells. The rest is covered by 20S and 30S. Authors are kindly asked to extend the description a bit detailed and more inclusive.
The description was extended in the introduction and additional references added.
General question: As authors mentioned in the intro that CDH1 helps APC/C to target proteins; but how CDH1 overexpression helps to increase the turnover rate of STAU1 and RBD2 domain while the protein level of APC/C complex remains unchanged? Do APC/C protein levels get changed upon CDH1 overexpression and/or upon different STAU1 wt and deletion constructs’ overexpression (Fig. 1D and 2E)? Have authors considered to include CDH1 knockdown condition besides to CDH1 overexpression?
APC/C is a complex E3 ligase composed of 13 subunits. It is inactive unless activated by complexes mechanisms that include members of the CDC20 family, pseudosubstrate inhibitors, protein kinases and phosphatases and the spindle assembly checkpoint. CDH1 overexpression is believed to activate APC/C.
Authors have mentioned only two sentences (Lines 61-64) about STAU1 (63kDa); then directly switched to isoform-2 STAU1 (55kDa) for the rest of the manuscript. When compared these two sequences, STAU1 (63kDa) is also containing the FPL-motif: What are the functional differences between these two variants? What is the scientific reasoning to particularly investigate the STAU1 (55kDa) but not both isoforms in the context of FPL-motif dependent turnover?
Very little is known about STAU1 (63 kDa). Indeed, almost all the studies to date were done with STAU1 (55 kDa), which is the most expressed of the two isoforms. In addition, although not directly studied, our previous paper on STAU1 degradation (Boulay et al., NAR 2014) suggests that STAU1 (63 kDa) behaves in the same way as STAU1 (55 kDa) for its degradation by APC/C during mitosis. Therefore, we did not want to duplicate all experiments. We added a short sentence in the introduction.
The “F37A” typo should be corrected as “Y37A” in the upper blot of Fig.3. We made the correction.
Authors are kindly asked to submit the raw mass spectrometry files to a public depository such as MassIVE or PRIDE.
Unfortunately, we do not have the raw data that were generated by a proteomic core facility. We received processed data. We will ask them to send the raw data to a public depository. Nevertheless, the complete processed data are included in the supplementary Tables linked to the paper.
What exactly makes the STAU1 but not MAP4K1 targeted by APC/C while both having the FPL-motif? What residues could be the determinant in the FPL-motif (or at some upstream/downstream residues of FPL-motif) to make it highly-targeted to certain
substrates? Or could that be subcellular restriction of the MAP4K1 localization that does not permit to be targeted by APC/C?
We believe that MAP4K1 is not degraded by APC/C because their subcellular localization is not the same. APC/C is a nuclear protein and is activated during mitosis. MAP4K1 is located in membranes and no report links this protein to mitosis. We added a sentence in the discussion.
A suggestion for the discussion: A recent study revealed that C-terminal glutamine residues could get targeted by TRIM family protein TRIM7 (PMID: 35982226). When the C-terminal glutamine residues (477-QQ-478) of STAU1 considered in that context, it would be nice to discuss the potential involvement of other E3 ligases, perhaps other TRIM family proteins in the turnover of STAU1.
Although QQ (477-478) are located in the C-terminal domain of STAU1, they are not C- terminal residues. It is thus not clear whether TRIM7 can recognize them or not.
Nevertheless, we added a brief (more general) discussion on the complexity of protein turnover via the involvement of multiple E3 ubiquitin ligases.
Round 2
Reviewer 1 Report
I am not satisfied with the revision. It is a major revision, explanations and arguments are not enough to make the conclusion reliable. Specifically, new data are required to answer the questions 2, 3, 5, 6, and 7. As for question 2, the difference between STAU1 and MAP4K1 in degradation is important for the FPL motif. Fig 5 showed no interaction between MAP4K1 and APC/C, questioning the role of FPL motif in the interaction between STAU1 and APC/C. Furthermore, the conclusion about FPL is a degradation motif is in doubt. Similarly, question 8 about the role of FPL in MAP4K1 degradation. More mutation is required for sure to confirm the conclusion.
Author Response
We thank the reviewers for their positive and constructive comments on the paper.
Reviewer 1:
Comments and Suggestions for Authors
I am not satisfied with the revision. It is a major revision, explanations and arguments are not enough to make the conclusion reliable. Specifically, new data are required to answer the questions 2, 3, 5, 6, and 7. As for question 2, the difference between STAU1 and MAP4K1 in degradation is important for the FPL motif. Fig 5 showed no interaction between MAP4K1 and APC/C, questioning the role of FPL motif in the interaction between STAU1 and APC/C. Furthermore, the conclusion about FPL is a degradation motif is in doubt. Similarly, question 8 about the role of FPL in MAP4K1 degradation.
More mutation is required for sure to confirm the conclusion.
Question 2: STAU1 contains FPL-motif and is degraded by APC/C and TRIM25. MAP4K1 also contains FPL-motif and is degraded by TRIM25, but not APC/C. This difference should be explained. Moreover, total 16 proteins contain FPL-motif. Are APC/C, TRIM25, or other E3 ligases responsible for their degradation? Answer these questions will enhance the importance of the FPL-motif.
Concerning question 2 and Fig 5, we do not understand the link that this reviewer establishes between the lack of interaction between MAP4K1 and APC/C and the doubt that it should bring regarding the interaction between STAU1 and APC/C. There are multiple examples where a degron is not recognized by a E3 ligase. For example, STAU1 contains a D-box that is a known and efficient target for APC/C but still the STAU1 D-box sequence is not functional (Boulay et al NAR 2014). This is why we looked for another degron in the current paper. We suggest that MAP4K1 is not degraded by APC/C because their subcellular localization is not the same. APC/C is a nuclear protein and is activated during mitosis. MAP4K1 is located in membranes and no report links this protein to mitosis. Alternatively, an unfavorable structural context around the FPL-motif (or other degrons) may explain the biological observation. Experimentally explaining the difference between STAU1 and MAP4K1 in degradation is not an easy task. Comparison of structural analysis (crystallography) of the FPL-degron of several proteins may clarify the situation.
In addition, the characterization of STAU1 degradation by APC/C was reported in our previous paper (Boulay et al NAR 2014). We show that the degron is located in the first 88 N-terminal amino acids, that it binds APC/C and that it is necessary for STAU1 degradation. Although we did not prove in the previous paper that the FPL- motif is the degron, we believe that we brought convincing evidence in the current paper.
Concerning the characterization of the other 14 proteins for degradation, it represents a long-term project outside the scope of this paper. There are more than 700 E3
ubiquitin ligases in human and identifying which one degrades the proteins may take some times.
- Total 8 Lys residues are in RBD2 domain. Mutation with deletion 37 residues in N- terminal (delta 37) of RBD2 is degraded by APC, suggesting Lys 2, 5, 6, 10 are not involved. The left 4 Lys are Lys60, 62, 68, and 74 are candidates for ubiquitination modification. It is interesting to know whether the distance between FPL-motif and Lys residues is essential for the regulation.
Proteomic analysis revealed that lys74 is ubiquitinated. However, many other lys residues (193, 199, 288, 292, 307, 324, 373, 432) are also ubiquitinated in STAU1. In MAP4K1, seven ubiquinated sites have been reported. It is not known how many of these lys residues are ubiquitinated after interaction with APC/C, TRIM25 or other E3 ligase. Defining the distance between the FPL-motif and the target Lys residues is not obvious since it must be analyzed in the context of 3D structure of the protein. In addition, defining which site(s) is ubiquitinated by each individual E3 ligase is a challenging and long-term experiment.
5 and 6. Fig 1D and Fig 2E should include overexpressing of CDC20 and APC/C. And importantly, the interaction resulting should be provided.
Overexpression of APC/C is a difficult experiment since APC/C is a complex E3 ligase composed of 13 subunits. Its activation is also highly regulated by different members of the CDC20 family, pseudosubstrate inhibitors, protein kinases and phosphatases and the spindle assembly checkpoint.
- Table 1 should provide the information similar to supplemental Fig 2B and C for all
This information was added in supplemental figures for all proteins (Supp Fig 2 D-M).
- Fig 4A, mutation causes less MAP4K1 F651A expressed in HEK293T Other important residues in FPL-motif should be selected for mutation assay.
Expression of WT and mutated MAP4K1 varies with transfection efficacy. They are expressed at the same levels in Fig 6 and slightly differently in other figures. We always managed to express low amounts of the proteins to avoid protein saturation and to optimize detection of even small variation in protein amounts when cell culture conditions are changing.
Reviewer 2 Report
Authors have clearly addressed all my questions and concerns.
Just two reminders: I would again emphasize that in the final version of the manuscript, authors should submit the raw mass spec data to a public available depository, that makes the data more transparent and further investigatable/analyzable by other scientists.
Another point: As authors mentioned about the lack of potential important negative controls in the proteomics analysis in authors response, I would strongly recommend a sentence in the discussion about potential false-positivity of some hits, particularly hits having low total spectral counts.
Author Response
We thank the reviewers for their positive and constructive comments on the paper.
Reviewer 2:
Comments and Suggestions for Authors
Authors have clearly addressed all my questions and concerns.
Just two reminders: I would again emphasize that in the final version of the manuscript, authors should submit the raw mass spec data to a public available depository, that makes the data more transparent and further investigatable/analyzable by other scientists.
Raw mass spec data have been deposited to DRIVE. A sentence was added in the paper to give the accession number.
Another point: As authors mentioned about the lack of potential important negative controls in the proteomics analysis in authors response, I would strongly recommend a sentence in the discussion about potential false-positivity of some hits, particularly hits having low total spectral counts.
My point is not that we did not use important negative controls in our analysis, but that optimal (perfect) negative controls are lacking for this technique. Indeed, we only analyzed proteins with total spectrum counts enriched at least four times compared to controls. A sentence was added in the manuscript.
Round 3
Reviewer 1 Report
I am not satisfied with the arguments. New experiments are required.
Author Response
My comments were included in the last round of evaluation.